# Synthesis of Mono- and Polyazole Hybrids Based on Polyfluoroflavones

**DOI:** 10.3390/molecules28020869

**Published:** 2023-01-15

**Authors:** Mariya A. Panova, Konstantin V. Shcherbakov, Ekaterina F. Zhilina, Yanina V. Burgart, Victor I. Saloutin

**Affiliations:** Postovsky Institute of Organic Synthesis, Ural Branch of the Russian Academy of Sciences, 22/20 Kovalevskoy St, Yekaterinburg 620108, Russia

**Keywords:** polyfluoroflavones, 1*H*-1,2,4-triazole, imidazole, nucleophilic aromatic substitution, regioselectivity, azolyl-substituted flavones, photoluminescence

## Abstract

The possibility of functionalization of 2-(polyfluorophenyl)-4*H*-chromen-4-ones, with them having different numbers of fluorine atoms, with 1,2,4-triazole or imidazole under conditions of base-promoted nucleophilic aromatic substitution has been shown**.** A high selectivity of mono-substitution was found with the use of an azole (1.5 equiv.)/NaOBu*^t^*(1.5 equiv.)/MeCN system. The structural features of fluorinated mono(azolyl)-substituted flavones in crystals were established using XRD analysis. The ability of penta- and tetrafluoroflavones to form persubstituted products with triazole under azole (6 equiv.)/NaOBu*^t^*(6 equiv.)/DMF conditions was found in contrast to similar transformations with imidazole. On the basis of mono(azolyl)-containing polyfluoroflavones in reactions with triazole and pyrazole, polynuclear hybrid compounds containing various azole fragments were obtained. For poly(pyrazolyl)-substituted flavones, green emission in the solid state under UV-irradiation was found, and for some derivatives, weak fungistatic activity was found.

## 1. Introduction

Flavones based on a 2-phenylchromen-4-one backbone are important heteroaromatic scaffolds in organic chemistry due to their availability and significant synthetic and biological potential [1,2,3,4,5]. The uniqueness of this heterocyclic backbone is also due to the fact that its derivatives are widely represented in the plant world, which often determines their diverse biological action [6,7,8,9,10,11]. Isolation, identification, and chemical modification of flavones of plant origin is one of the rapidly developing areas in drug design [12,13]. Another equally important area of progress in the chemistry of flavones is the development of synthetic strategies for their modification. In addition, here, certain successes have recently been achieved, for example, its modification has been proposed by the Buchwald–Hartwig reaction [14], metal-catalyzed cascade rearrangements [15,16], electrochemical dimerization [17], CH-functionalization [18], etc. However, all of these transformations most often require expensive catalysts or complex installations.

In turn, fluorinated flavones offer extra possibilities for their functionalization in reactions with nucleophilic reagents. Fluoroaromatic compounds are well known to be perspective frameworks for their modification by different methods for formation of new C–C and C–heteroatom bonds [19,20,21], including C–N bond formation in reactions with azole-type heterocycles [22,23,24]. There are known approaches to the synthesis of chromone–azole dyads [25]; however, data on direct functionalization of fluorine-containing flavones with azoles under S_N_Ar reaction conditions have only been only in publications from our research team [26,27], although the reaction of nucleophilic aromatic substitution of fluorine atoms is a quite simple, economically and environmentally friendly process, which offers the possibility of substitution for fluorinated substrates with advantages over reactions that use expensive catalysts [28]. To date, we have considerable practice in the synthesis and modification of polyfluoroflavones [26,27,29,30,31,32]. Previously, we proposed a convenient and efficient method for the synthesis of B-ring polyfluorinated flavones (2-(polyfluorophenyl)chromen-4-ones) [29], which can be involved in S_N_Ar reactions to obtain polynuclear heterocyclic compounds based on flavones and azoles. We have shown the possibility of controlling the fluorine atoms substitution of 2-(polyfluorophenyl)chromen-4-ones by pyrazole and assumed the mechanism for sequential fluorine substitution on an example of 2-pentafluorophenyl-4*H*-chromen-4-one [26].

Within this work in continuation of our research in this area, the features of the functionalization of 2-(polyfluoroaryl)-4*H*-chromen-4-ones **1**–**3**, with them having different numbers of fluorine atoms in the aryl substituent (Figure 1), by 1*H*-1,2,4-triazole and imidazole under conditions of base-promoted nucleophilic aromatic substitution were studied.

The introduction of azole fragments has a wide perspective due to their ability to form various noncovalent interactions with different therapeutic targets, which is valuable for drug design. Different azole derivatives have significant potential for medicinal chemistry [33,34,35,36,37,38,39,40]. Of particular interest are 1*H*-1,2,4-triazole and imidazole derivatives, which are known to possess therapeutic effect against drug-resistant pathogens [40,41,42]. The way to more effective medicines is through the synthesis of azole hybrids with other pharmacophores [27,43,44,45], which might be flavones, with them having a pyran framework that determines their great potential as antiviral antibacterial agents. Polyazole hybrid derivatives are also given special attention due to their potential applications as electron-transporting materials, emitters, and host materials in OLEDs, the most attractive products of organic electronics [46,47,48]. In this regard, the synthesis of hybrid compounds based on the flavone and azole cycles seems to be a prominent problem, which can be solved by the S_N_Ar reaction of polyfluorinated flavones with azoles.

## 2. Results

The study of the reaction of polyfluoroflavones **1–3** with 1*H*-1,2,4-triazole and imidazole under base-promoted nucleophilic aromatic substitution was carried out according to three synthetic protocols developed during the research of transformations with pyrazole [26]. The application of the Cs_2_CO_3_-promoted conditions will allow for observation of the spectrum of possible substituted products, while the application of the NaOBu*^t^*-promoted conditions should facilitate for selective mono- and persubstitution of fluorine, which depends on variation of the nucleophile and base loading. The most convenient method for identifying fluorine-containing compounds in their mixtures is ^19^F NMR spectroscopy data.

The reaction of flavone **1** with 3 equiv. of triazole and Cs_2_CO_3_ in MeCN led to a mono-, tri-, tetra-, and penta(1*H*-1,2,4-triazol-1-yl)-substituted products mixture, from which only mono- and penta-substituted flavones **4** and **7** were isolated by column chromatography (Figure 1). The ^19^F NMR data of fluorine-containing products **4**–**7** and their ratio in the reaction mixture are shown in Table 1. Under optimized conditions, selective synthesis of mono- and penta(1*H*-1,2,4-triazol-1-yl)-substituted products **4** and **7** with a good preparative yield was achieved (Figure 1).

In contrast to the transformations with triazole, the reaction of flavone **1** with 3 equiv. of imidazole under the same conditions resulted in the formation of mono-, di-, and tri(1*H*-imidazol-1-yl)-substituted products **8**–**10**, which were isolated with a poor preparative yield. The ^19^F NMR data of fluorine-containing products **8**–**10** and their ratio in the mixture are shown in Table 2. Under conditions preferable to monosubstitution, 2-[2,3,5,6-tetrafluoro-4-(1*H*-imidazol-1-yl)phenyl]-4*H*-chromen-4-one **8** was synthesized with a good yield (Figure 2). In addition, under conditions conducive to the formation of a persubstituted product, the reaction of these reagents is extremely nonselective.

It should be noted that for 2-[2,5-difluoro-3,4,6-tri(1*H*-imidazol-1-yl)phenyl]-4*H*-chromen-4-one **10**, an isomeric structure—2-[3,5-difluoro-2,4,6-tri(1*H*-imidazol-1-yl)phenyl]-substituted product can be proposed. The structures of **9** and **10** were solved on the basis of a comparative analysis of ^1^H, ^19^F NMR data of these compounds **9** and **10,** and NMR data of formerly synthesized [26] tri(1*H*-pyrazol-1-yl)-substituted analogue **A** (Appendix B, Table A1).

The crystal structures of 2-[2,3,5,6-tetrafluoro-4-(1*H*-azol-1-yl)phenyl]-4*H*-chromen-4-ones flavones **4** and **8** were confirmed by XRD analysis (Figure 2 and Figure 3). Both homologues **4** and **8** have similar structural characteristics, in contrast to the pyrazolyl-substituted analogue [26], which does not have intramolecular interactions coordinating both pyrone and aryl moieties, which are therefore aplanar. In addition, the introduction of triazole and imidazole fragments has a critical effect on the structure of their unit cell in the crystal. Thus, a cell of compound **4** has a rhombic syngony and compound **8** has a monoclinic syngony. It is important to note that for two systems of atoms C16C11C2C3 and C17N1C14C15 of product **4** and similar systems C3C2C11C12 and C13C14N1C17 of analogue **8**, the torsion angles have values close in absolute value, but pairwise opposite in sign, equal to 42.70(0.76), −47.50(0.85), and −58.89(0.31), 56.50(0.34) degrees for **4** and **8,** respectively. The unit cell of the crystal **4** consists of four molecules due to the formation of O**^…^**H, F**^…^**F, and C**^…^**C sp2 short intermolecular contacts (O3**^…^**H17 2.373, O3**^…^**H18 2.634, C5**^…^**C14 3.331(8), C9**^…^**C12 3.303(7), F2**^…^**F4 2.889(5) Å) (Figure 2). The unit cell of crystal **8** also consists of four molecules stabilized by C**^…^**H, C**^…^**F, and C**^…^**C sp2 short intermolecular contacts (C18**^…^**H5 2.80(2), C7**^…^**F3 3.117(3), C9**^…^**C16 3.396(3) Å) (Figure 3).

Furthermore, we introduced flavone **2**, with it having 2,3,4,5-tetrafluorophenyl substituent, in the S_N_Ar reaction with triazole and imidazole. The reaction of compound **2** with 3 equiv. of triazole (Figure 3) and imidazole (Figure 4) in the presence of Cs_2_CO_3_ in MeCN led to mono-, di-, and tri(azol-1-yl)-substituted products mixture **11**–**13** and **15**–**17**, respectively. The ^19^F NMR data of fluorine-containing flavones **11**–**13** and **15**–**17** and their ratio in the mixture are shown in Table 3. Under optimized conditions, selective synthesis of mono- **11** and tetra(triazol-1-yl)-substituted products **14** was conducted with a good preparative yield (Figure 3). However, under similar conditions, only mono(imidazolyl)-substituted flavone **15** was obtained in a good yield from the reaction with imidazole, and it was not possible to isolate the persubstituted product (Figure 4) as in the reaction of flavone **1** (Figure 2). This may indicate a lower reactivity of imidazole compared to triazole and the previously studied pyrazole under the conditions used [26].

The structure of 2-[2,3,5-trifluoro-4-(1*H*-imidazol-1-yl)phenyl]-4*H*-chromen-4-one **15** was confirmed by XRD analysis (Figure 4). As well as products **4** and **8**, compound **15** has no intramolecular interactions, which coordinated both pyrone and aryl moieties. However, the replacement of fluorine at the C6′ site by hydrogen allows the mutual position of these two moieties of flavone **15** in crystal close to coplanar. Torsions C3C2C11C12 and C1N1C14C13 are −7.10(0.30) and −48.14(0.27) degrees. The unit cell of the crystal **15** is monoclinic, consists of four molecules, stabilized by pairs of O**^…^**H, C**^…^**O short intermolecular contacts (O4**^…^**H17 2.33(3), C17**^…^**O4 3.148(3), and C2**^…^**O4 3.127(2) Å).

The reaction of 2-(3,4,5-trifluoro-2-methoxy-phenyl)-4*H*-chromen-4-one **3** with 3 equiv. of triazole in the presence of Cs_2_CO_3_ in MeCN led to the formation of mono- and di(1*H*-1,2,4-triazol-1-yl)-substituted products **18**, **19**, which were isolated from the mixture (Figure 5). In the ^19^F NMR spectrum of the mixture, in addition to the signals corresponding to products **18** and **19**, two pairs of signals were recorded, presumably assigned by us to compounds **20** and **21** (Table 4). Due to the low intensity and insufficient resolution of these signals, the possibility of their detection is difficult. We believe that the formation of product **20** is possible due to the demethylation of flavone **18** under the reaction conditions used. The formed phenolic group in compound **20** can be in equilibrium between keto and enol forms, and keto form may undergo a nucleophilic addition reaction with triazole followed by aromatization to provide flavone **21**.

Under NaOBu*^t^*-promoted conditions for selective monosubstitution corresponding 2-[2,3,5,6-tetrafluoro-4-(1*H*-1,2,4-triazol-1-yl)phenyl]-4*H*-chromen-4-one **18** was synthesized with a good preparative yield. The reaction with four-fold excess of triazole and NaOBu*^t^* led to flavone **19** and 2-[2-hydroxy-3,4-di(1*H*-1,2,4-triazol-1-yl)phenyl]-4*H*-chromen-4-one **22**, which was obtained by demethylation of 2-methoxy-3,4-di(1*H*-1,2,4-triazol-1-yl)-substituted precursor **19**, with poor yields, similar to pyrazolyl-substituted analogues [26]. The use of six-fold excess of triazole and NaOBu*^t^* gave the same result as above with four-fold excess.

From the reaction of flavone **3** with 3 equiv. of imidazole in the presence of Cs_2_CO_3_ in MeCN, only mono-substituted product **23** was isolated individually. The ^19^F NMR spectrum of the mixture (Table 5) also contained signals assigned to flavones **24** and **25** based on the conclusions given above for similar transformations with triazole. Under optimized conditions, 2-[3,5-difluoro-2-methoxy-4-(1*H*-imidazol-1-yl)phenyl]-4*H*-chromen-4-one **23** was synthesized (Figure 6). Using an excess of this azole has not been successful in isolating any individual products.

The structures of 2-[5-fluoro-2-methoxy-3,4-di(1*H*-1,2,4-triazol-1-yl)phenyl]-4*H*-chromen-4-one **19** and 2-[4,5-difluoro-2-methoxy-4-(1*H*-1,2,4-imidazol-1-yl)phenyl]-4*H*-chromen-4-one **23** were confirmed by XRD analysis (Figure 5 and Figure 6). Compounds **19** and **23** also do not have any intramolecular interactions. It was found that the introduction of one or two azole fragments affects the geometric parameters of crystals **19** and **23**. The unit cell of compound **19** has a triclinic syngony and consists of two molecules stabilized by a pair of short intramolecular contacts C**^…^**F (C10**^…^**F1 3.159(6) Å). The cell of compound **23** of the monoclinic system consists of four molecules forming short intermolecular contacts O**^…^**H, C**^…^**F (O3**^…^**H8 2.568, O3**^…^**H16 2.32(3), C3**^…^**F001 3.148(3) Å). It should be noted that the torsion angles of two similar systems of atoms C16C11C2C3 and C3C2C11C12 of di(triazolyl)- and mono(imidazolyl)-substituted flavones **19** and **23** have a significant difference both in absolute value and in sign and are equal to −22.81(0.72) and 8.64(0.33) degrees, respectively. The geometry of the azole fragments of molecules **19** and **23** does not reveal fundamental differences. The torsion angles of the atomic systems C18N1C15C16, C20N4C14C15, and C17N1C14C13 are 56.12(0.58), 56.96(3.50), and 49.82(0.37) degrees, respectively.

It should be noted that a common characteristic of the crystals of both mono(1*H*-imidazol-1-yl)-substituted products **15** and **23** is the formation of a unit cell of the monoclinic system. However, the torsion angles of the systems of atoms C3C2C11C12, C1N1C14C13 of trifluoroflavone **15** and C3C2C11C12 and C17N1C14C13 of difluoroflavone **23** have practically equal absolute values and opposite values (Figure 4 and Figure 6).

In this work, we also studied the possibility of the synthesis of 2-[poly(1*H*-azol-1-yl)phenyl]-4*H*-chromen-4-ones, involving two different azole-type heterocycles in their structure by the NaOBu*^t^*-promoted S_N_Ar reaction of mono(1*H*-azol-1-yl)-substituted flavones with azoles.

Thus, the reaction of mono(triazolyl)-substituted flavone **4** with four-fold excess of pyrazole and NaOBu*^t^* led to 2-[2,5-difluoro-3,6-di(1*H*-pyrazol-1-yl)-4-(1*H*-1,2,4-triazol-1-yl)phenyl]- and 2-[2,3,5,6-tetra(1*H*-pyrazol-1-yl)-4-(1*H*-1,2,4-triazol-1-yl)phenyl]-4*H*-chromen-4-ones **26** and **27**. Flavone **26** was shown to react with two-fold excess of pyrazole and NaOBu*^t^* to form product of complete substitution **27** (Figure 7). Under conditions preferable for persubstitution, flavone **8** reacts with pyrazole and triazole, leading to the formation of compounds **28** and **29**. Similarly, 2-(2,3,5,6-tetrafluoro-4-(1*H*-pyrazol-1-yl)phenyl)-4*H*-chromen-4-one **30** obtained earlier [26] forms product **31** with triazole (Figure 8). Unfortunately, we have not yet succeeded in growing a suitable single crystal for XRD due to the limited solubility of polynuclear products **27**–**29** and **31** in organic solvents.

As is known, aryl-azole scaffolds are considered as promising materials for OLEDs [46,47,48,49]; therefore, we recorded the photoluminescence spectra for poly(azolyl)-substituted flavones **28**–**31** and formerly synthesized penta(pyrazolyl)-substituted analogue **32** [26]. It was found that 2-[4-(imidazolyl)-2,3,5,6-tetra(pyrazolyl)- and penta(pyrazolyl)-substituted flavones **28** and **32** exhibit emission in the solid state under UV-irradiation, in contrast to poly(triazolyl)-containing derivatives **29** and **31**. The emission spectra of these compounds were recorded, and the data are presented in Table 6 and Figure 7. Flavones **28** and **32** possess green emission with maxima 504 nm. The commission international de L’Eclairage (CIE) coordinates were (0.255; 0.528) and (0.255; 0.526) for **28** and **32**, respectively. Substitution of pyrazole fragment at C4′ by imidazole results in a 1.6-fold increase in quantum yield (0.18 and 0.29 for **28** and **32**, correspondingly). Detailed data of the fluorescence lifetime measurements of flavones **28** and **32** are given in Appendix C.

Azolyl-substituted flavones are of great interest in the search for the bioactive compounds among them, and, in particular, for antimycotic agents [50]; therefore, we screened the fungistatic activity of a number of compounds (Appendix D) obtained in this work and earlier [29] in relation to four control strains of clinically significant species of pathogenic fungi *Trichophyton rubrum, Epidermophyton floccosum, Microsporum canis,* and *Candida parapsilosis.* It was found that flavones **8** and **14** have a weak inhibitory effect against *T. rubrum* and *M. canis* (MIC 100 mg/mL), and derivatives **23** and **33**, combined methoxy and azole substituents, in the absence of fungistatic activity (MIC > 100 µg/mL) showed high and moderate activity in inhibiting the growth of 50% fungal culture (MIC_50_ 1.56–12.5 µg/mL).

## 3. Materials and Methods

### 3.1. Chemistry: General Information and Synthetic Techniques

Solvents and reagents except fluorine-containing flavones are commercially available and were used without purification. The NMR spectra of the synthesized compounds (see Appendix A) were recorded on Bruker DRX-400 and Bruker AVANCE III 500 spectrometers (^1^H, 400.13 (DRX400) and 500.13 (AV500) MHz, ^13^C, 125.76 MHz, Me_4_Si as an internal standard, ^19^F, 376.44 (DRX400) and 470.52 (AV500) MHz, C_6_F_6_ as an internal standard, chemical shifts were not converted to CCl_3_F)). IR spectra were recorded on a Perkin Elmer Spectrum Two FT-IR spectrometer (UATR) in the range of 4000–400 cm^–1^. Elemental (C, H, N) analysis was performed on a Perkin Elmer PE 2400 Series II CHNS-O EA 1108 elemental analyzer. The melting points were measured on a Stuart SMP3 in open capillaries. The reaction progress was monitored by TLC on ALUGRAM Xtra SIL G/UV_254_ sheets. The starting flavones (**1**–**3**) were synthesized by a procedure [29].

Synthetic technique A for the synthesis of azolyl-substituted flavones using Cs_2_CO_3_. flavone (0.5 or 1 mmol), azole (3 equiv.), and Cs_2_CO_3_ (3 equiv.) were suspended in 10 mL of MeCN. The reaction mixture was heated to 80 °C. The reaction progress was monitored by TLC. At the end of the reaction, the mixture was diluted with water (10 mL) and extracted with CHCl_3_ or DCM (2 × 10 mL). Organic layers were combined, and the solvent was removed. The residue was immobilized on silica gel and purified by column chromatography using an appropriate eluent mixture (2:1 *v*/*v*).

Synthetic technique B for the synthesis of azolyl-substituted flavones using NaOBu*^t^*. flavone (0.5 or 1 mmol) was dissolved in dry DMF (5 mL), placed in a sealed vial and cooled to 0 °C. Azole (from 1.5 to 6 equiv.) and NaOBu*^t^* (from 1.5 to 6 equiv.) were suspended in dry DMF (5 mL) and stirred at room temperature for 5 min. The mixture was cooled to 0 °C and was added to a flavone solution in DMF while stirring. After 10 min. in a cooling bath (0 °C), the vialwas removed and the reaction continued at room temperature. The reaction progress was monitored by TLC. At the end, the reaction mixture was diluted with water (10 mL), stirred, and cooled. The formed precipitate was filtered off and washed with water. The water solution was neutralized with 0.1M HCl and extracted with CHCl_3_ or DCM. The organic layer was separated, and the solvent was removed. Organic residues were combined, immobilized on silica gel and purified by column chromatography using an appropriate eluent mixture (2:1 *v*/*v*).

### 3.2. Spectral and Elemental Analysis Data of Synthesized Compounds

2-[2,3,5,6-Tetrafluoro-4-(1*H*-1,2,4-triazol-1-yl)phenyl]-4*H*-chromen-4-one (**4**). Yield 202 mg (56% according to technique A), 256 mg (71% according to technique B); white powder; mp 192–194 °C; IR *ν* 3108, 3043 (C–H^Ar^), 1639 (C=O), 1528, 1485, 1381 (C=C^Ar^, C–H^Ar^, C–N), 1139, 1110 (C–F) cm^−1^; ^1^H NMR (500.13 MHz, CDCl_3_) *δ* 6.68 (s, 1H, CH^Pyranone^), 7.50 (m, 1H, CH^Ar^), 7.53 (d, *J* = 8.6 Hz, 1H, CH^Ar^), 7.76 (ddd, *J* = 8.6, 7.1, 1.7 Hz, 1H, CH^Ar^), 8.26 (dd, *J* = 8.0, 1.5 Hz, 1H, CH^Ar^), 8.29 (s, 1H, CH^Triazole^), 8.52 (s, 1H, CH^Triazole^) ppm; ^13^C NMR (125.76 MHz, CDCl_3_) *δ* 113.7 (t, *J* = 15 Hz, C^Ar^), 116.1 (t, *J* = 3 Hz, C^Ar^), 118.2 (s, C^Ar^), 118.8 (s, C^Ar^), 123.9 (s, C^Pyranone^), 125.9 (s, C^Ar^), 126.0 (s, C^Ar^), 134.5 (s, C^Ar^), 141.7 (m, 2C^ArF^), 144.7 (ddt, *J* = 257, 14, 5 Hz, 2C^ArF^), 145.3 (t, *J* = 3 Hz, C^Triazole^), 151.7 (s, C^Ar^), 153.5 (s, C^Triazole^), 156.6 (s, C^Pyranone^), 177.0 (s, C^Pyranone^) ppm; ^19^F NMR (470.52 MHz, CDCl_3_) *δ* 16.66 (m, 2F), 24.83 (m, 2F) ppm; Anal. calcd. for C_17_H_7_F_4_N_3_O_2_: C 56.52, H 1.95, N 11.63, found: C 56.26, H 1.84, N 11.38.

2-[2,3,4,5,6-Penta(1*H*-1,2,4-triazol-1-yl)phenyl]-4*H*-chromen-4-one (**7**). Yield 39 mg (7% according to technique A), 335 mg (60% according to technique B); light-yellow powder; mp 322–323 °C; IR *ν* 3118, 3086 (C–H^Ar^), 1651 (C=O), 1514, 1459, 1379, 1272 (C=C^Ar^, C–H^Ar^, C–N) cm^−1^; ^1^H NMR (500.13 MHz, CDCl_3_) *δ* 6.20 (s, 1H, CH^Pyranone^), 7.16 (d, *J* = 8.4 Hz, 1H, CH^Ar^), 7.42 (m, 1H, CH^Ar^), 7.65 (ddd, *J* = 8.5, 7.2, 1.7 Hz, 1H, CH^Ar^), 7.87 (s, 1H, CH^Triazole^), 7.88 (s, 2H, CH^Triazole^), 7.90 (s, 2H, CH^Triazole^), 8.08 (dd, *J* = 8, 1.6 Hz, 1H, CH^Ar^), 8.10 (s, 1H, CH^Triazole^), 8.15 (s, 2H, CH^Triazole^), 8.22 (s, 2H, CH^Triazole^) ppm; ^13^C NMR (125.76 MHz, CDCl_3_) *δ* 114.9, 117.4 (s, 2C^Ar^), 123.2, 126.1, 126.5, 133.1, 134.3 (s, 2C^Ar^), 134.9, 135.3, 136.1, 145.3 (s, 2C^Triazole^), 145.7, 146.0 (s, 2C^Triazole^), 153.4, 153.7 (s, 2C^Triazole^), 153.7 (s, 2C^Triazole^), 155.7, 175.8 ppm; Anal. Calcd. For C_25_H_15_N_15_O_2_: C 53.86, 2.71, N 37.69, found: C 53.60, H 2.68, N 37.69.

2-[2,3,5,6-Tetrafluoro-4-(1*H*-imidazol-1-yl)phenyl]-4*H*-chromen-4-one (**8**). Yield 137 mg (38% according to technique A), 263 mg (73% according to technique B); white powder, mp 172–174 °C; IR *ν* 3110 (C–H^Ar^), 1653 (C=O)*,* 1489, 1459, 1376 (C=C^Ar^, C–H^Ar^, C–N), 1224 (C–F) cm^−1^; ^1^H NMR (400.13 MHz, CDCl_3_) *δ* 6.67 (s, 1H, CH^Pyranone^), 7.31 (br.s., 1H, CH^Imidazole^), 7.33 (br.s., 1H, CH^Imidazole^), 7.48–7.53 (m, 2H, 2CH^Ar^), 7.76 (ddd, *J* = 8.6, 7.3, 1.5 Hz, 1H, CH^Ar^), 7.88 (br.s., 1H, CH^Imidazole^), 8.27 (dd, *J* = 7.9, 1.5 Hz, 1H, CH^Ar^) ppm; ^13^C NMR (125.76 MHz, CDCl_3_) *δ* 112.0 (t, *J* = 15 Hz, C^Ar^), 115.9 (t, *J* = 3 Hz, C^Ar^), 118.2, 119.3 (t, *J* = 13 Hz, C^Ar^), 119.7 (t, *J* = 2 Hz, C^Imidazole^), 123.9, 125.9, 126.0, 130.6, 134.5, 137.5 (t, *J* = 4 Hz, C^Imidazole^), 140.1–142.3 (m, 2C^ArF^), 144.9 (ddt, *J* = 257, 14, 5 Hz, 2C^ArF^), 152.0, 156.6, 177.1 ppm; ^19^F NMR (376.44 MHz, CDCl_3_) *δ* 14.88 (dqd, *J* = 6.0, 4.4, 2.4 Hz, 2F), 24.51 (dq, *J* = 7.3, 4.2 Hz, 2F) ppm; Anal. calcd. for C_18_H_8_F_4_N_2_O_2_: C 60.01, H 2.24, N 7.78, found: C 59.89, H 2.30, N 7.96.

2-[2,3,5-Trifluoro-4,6-di(1*H*-imidazol-1-yl)phenyl]-4*H*-chromen-4-one (**9**). Yield 20 mg (5%); light-yellow powder, mp 200–203 °C; ^1^H NMR (400.13 MHz, CDCl_3_) *δ* 6.52 (d, *J* = 1.5 Hz, 1H, CH^Pyranone^), 7.05 (br.s., 1H, CH^Imidazole^), 7.13 (br.s., 1H, CH^Imidazole^), 7.18 (d, *J* = 8.4 Hz, 1H, CH^Ar^), 7.33 (br.s., 1H, CH^Imidazole^), 7.35 (br.s., 1H, CH^Imidazole^), 7.41–7.45 (m, 1H, CH^Ar^), 7.65 (br.s., 1H, CH^Imidazole^), 7.65 (dd, *J* = 15.7, 1.7 Hz, 1H, CH^Ar^), 7.90 (br.s., 1H, CH^Imidazole^), 8.17 (dd, *J* = 7.9, 1.5 Hz, 1H, CH^Ar^) ppm; ^19^F NMR (376.44 MHz, CDCl_3_) *δ* 23.60 (d, *J* = 22.2 Hz, 1F), 26.33 (ddd, *J* = 22.6, 13.0, 1.3 Hz, 1F), 30.57 (d, *J* = 13.0 Hz, 1F) ppm. Anal. calcd. for C_21_H_11_F_3_N_4_O_2_: C 61.77, H 2.72, N 13.72, found: C 61.89, H 2.79, N 13.66.

2-[2,5-Difluoro-3,4,6-tri(1*H*-imidazol-1-yl)phenyl]-4*H*-chromen-4-one (**10**). Yield 95 mg (17%); light-yellow powder; mp 227–228 °C; ^1^H NMR (400.13 MHz, CDCl_3_) *δ* 6.51 (d, *J* = 1.3 Hz, 1H, CH^Pyranone^), 6.81 (d, *J* = 1.3 Hz, 1H, CH^Imidazole^), 6.83 (d, *J* = 1.2 Hz, 1H, CH^Imidazole^), 7.10 (d, *J* = 1.2 Hz, 1H, CH^Imidazole^), 7.15–7.20 (m, 1H, CH^Ar^), 7.21–7.23 (m, 3H, 3CH^Imidazole^), 7.43–7.47 (m, 1H, CH^Ar^), 7.50 (br.s, 2H, 2CH^Imidazole^), 7.65–7.69 (m, 1H, CH^Ar^), 7.70 (br.s, 1H, 1CH^Imidazole^), 8.18 (dd, *J* = 8.0, 1.6 Hz, 1H, CH^Ar^) ppm; ^13^C NMR (125.76 MHz, CDCl_3_) *δ* 115.8 (d, *J* = 3 Hz, CH^Pyranone^), 117.8, 119.0, 119.2, 120.0, 120.5 (d, *J* = 17 Hz, C^ArF^), 123.3 (d, *J* = 16 Hz, C^ArF^), 123.6, 125.6 (dd, *J* = 15, 2 Hz, C^ArF^), 125.8–126.0 (m, C^ArF^), 125.9, 126.2, 131.0, 131.46, 131.47, 134.7, 136.9 (d, *J* = 3 Hz, C^ArF^), 137.0 (d, *J* = 2 Hz, C^Imidazole^), 137.5 (d, *J* = 2 Hz, C^Imidazole^), 148.3 (dd, *J* = 256, 4 Hz, C^ArF^), 150.9 (dd, *J* = 257, 4 Hz, C^ArF^), 152.7 (d, *J* = 2 Hz, C^Imidazole^), 156.2, 176.6 ppm; ^19^F NMR (376.44 MHz, CDCl_3_) *δ* 31.47 (dm, *J* = 14.4 Hz, 1F), 41.18 (dd, *J* = 14.7, 1.2 Hz, 1F) ppm. Anal. calcd. for C_24_H_14_F_2_N_6_O_2_: C 63.16, H 3.09, N 18.41, found: C 62.90, H 2.82, N 18.24.

2-[2,3,5-Trifluoro-4-(1*H*-1,2,4-triazol-1-yl)phenyl]-4*H*-chromen-4-one (**11**). Yield 257 mg (75%); white powder; mp 210–211 °C; IR *ν* 3138, 3118, 3070 (C–H^Ar^), 1634 (C=O), 1531, 1462, 1369 (C=C^Ar^, C–H^Ar^, C–N), 1040 (C–F) cm^−1^; ^1^H NMR (500.13 MHz, CDCl_3_) *δ* 7.02 (s, 1H, CH^Pyranone^), 7.49 (t, *J* = 7.5 Hz, 1H, CH^Ar^), 7.58 (d, *J* = 8.4 Hz, 1H, CH^Ar^), 7.74–7.79 (m, 2H, 2CH^Ar^), 8.25 (dd, *J* = 8.0, 1.6 Hz, 1H, CH^Ar^), 8.27 (s, 1H, CH^Triazole^), 8.49 (s, 1H, CH^Triazole^) ppm; ^13^C NMR (125.76 MHz, CDCl_3_) *δ* 110.6 (dd, *J* = 24, 4 Hz, C^Ar^), 113.9, 114.0, 118.0 (m, C^Ar^, C^Triazole^), 122.7 (t, *J* = 9 Hz, C^Ar^), 123.8, 125.9, 134.5, 144.8–145.1 (m, C^ArF^), 145.3 (t, *J* = 3 Hz, C^Triazole^), 146.9–147.2 (m, C^ArF^), 151.7 (ddd, *J* = 254, 3, 2 Hz, C^ArF^), 153.3, 154.9 (td, *J* = 4, 2 Hz, C^Pyranone^), 156.1, 177.6 ppm; ^19^F NMR (470.52 MHz, CDCl_3_) *δ* 23.72 (d, *J* = 20.1 Hz, 1F), 24.92 (ddd, *J* = 20.3, 14.5, 5.9 Hz, 1F), 39.24 (dd, *J* = 14.6, 10.8 Hz, 1F) ppm; Anal. calcd. for C_17_H_8_F_3_N_3_O_2_: C 59.48, H 2.35, N 12.24, found: C 54.40, H 2.32, N 12.29.

2-[2,3,4,5-Tetra(1*H*-1,2,4-triazol-1-yl)phenyl]-4*H*-chromen-4-one (**14**). Yield 333 mg (68%); light-yellow powder; mp 295–296 °C; ; IR *ν* 3134, 3108 (C–H^Ar^), 1644 (C=O), 1508, 1467 (C=C^Ar^, C–H^Ar^, C–N) cm^−1^; ^1^H NMR (500.13 MHz, (CD_3_)_2_SO) *δ* 6.87 (s, 1H, CH^Pyranone^), 7.22 (d, *J* = 8.3 Hz, 1H, CH^Ar^), 7.49–7.53 (m, 1H, CH^Ar^), 7.80 (ddd, *J* = 8.7, 7.4, 1.6 Hz, 1H, CH^Ar^), 8.00 (s, 1H, CH^Triazole^), 8.03 (dd, *J* = 7.9, 1.5 Hz, 1H, CH^Ar^), 8.05 (s, 1H, CH^Triazole^), 8.10 (s, 1H, CH^Triazole^), 8.17 (s, 1H, CH^Triazole^), 8.61 (s, 1H, CH^Triazole^), 8.68 (s, 1H, CH^Triazole^), 8.86 (s, 1H, CH^Triazole^), 8.87 (s, 1H, CH^Triazole^), 9.04 (s, 1H, CH^Triazole^) ppm; ^13^C NMR (125.76 MHz, CDCl_3_) *δ* 112.8, 118.1 (s, 2C^Ar^), 122.9, 124.9, 126.1, 129.3, 131.1, 132.9 (d, *J* = 2 Hz, C^Ar^), 133.6, 134.9, 135.2, 146.0, 146.8, 146.9, 147.2, 152.6, 152.7, 152.8, 153.1, 155.4, 158.6, 176.5 ppm; Anal. calcd. for C_23_H_14_N_12_O_2_: C 56.33, H 2.88, N 34.27, found: C 56.11, H 2.90, N 34.22.

2-[2,3,5-Trifluoro-4-(1*H*-imidazol-1-yl)phenyl]-4*H*-chromen-4-one (**15**). Yield 277 mg (81%); white powder; mp 204–206 °C; IR *ν* 3160, 3133, 3077 (C–H^Ar^), 1625, 1606 (C=O), 1515, 1465, 1351 (C=C^Ar^, C–H^Ar^, C–N), 1007 (C–F) cm^−1^; ^1^H NMR (500.13 MHz, CDCl_3_) *δ* 7.00 (s, 1H, CH^Pyranone^), 7.31 (br.s., 2H, 2CH^Imidazole^), 7.47–7.50 (m, 1H, 1CH^Ar^), 7.58 (d, *J* = 8.2 Hz, 1H, CH^Ar^), 7.71–7.74 (m, 1H, 1CH^Ar^), 7.75–7.78 (m, 1H, 1CH^Ar^), 7.88 (br.s., 1H, CH^Imidazole^), 8.25 (dd, *J* = 8.0, 1.6 Hz, 1H, CH^Ar^) ppm; ^13^C NMR (125.76 MHz, CDCl_3_) *δ* 110.6 (dd, *J* = 25, 4 Hz, C^Ar^), 113.5, 113.6, 118.0, 118.5 (dd, *J* = 17, 12 Hz, C^ArF^), 119.8 (t, *J* = 2 Hz, C^Imidazole^), 120.7 (t, *J* = 9 Hz, C^ArF^), 123.7, 125.9, 130.2, 134.4, 137.6 (t, *J* = 4 Hz, C^Imidazole^), 145.3 (ddd, *J* = 256, 17, 4 Hz, C^ArF^), 146.2 (ddd, *J* = 258, 14, 4 Hz, C^ArF^), 151.2 (dt, *J* = 251, 3 Hz, C^ArF^), 155.1 (m, C^ArF^), 156.1, 177.7 ppm; ^19^F NMR (470.52 MHz, CDCl_3_) *δ* 21.78 (dd, *J* = 19.8, 1.2 Hz, 1F), 24.83 (ddd, *J* = 20.1, 14.3, 1.2 Hz, 1F), 38.13–38.18 (m, 1F) ppm; Anal. calcd. for C_18_H_9_F_3_N_2_O_2_: C 63.16, H 2.65, N 8.18, found: C 63.16, H 2.81, N 8.29.

2-[5-Fluoro-2,3,4-tri(1*H*-imidazol-1-yl)phenyl]-4*H*-chromen-4-one (**17**). Yield 31 mg (14%); yellow powder; mp 168–170 °C; ^1^H NMR (400.13 MHz, CDCl_3_) *δ* 6.40 (s, 1H, CH^Pyranone^), 6.57 (t, *J* = 1.3 Hz, 1H, CH^Imidazole^), 6.73 (d, *J* = 1.1 Hz, 1H, CH^Ar^), 6.78 (t, *J* = 1.3 Hz, 1H, CH^Imidazole^), 7.03–7.04 (m, 2H, CH^Imidazole^), 7.13 (br.s, 1H, CH^Imidazole^), 7.15–7.17 (m, 2H, CH^Imidazole^), 7.39 (br.s, 1H, CH^Imidazole^), 7.40–7.44 (m, 1H, CH^Ar^), 7.47 (br.s, 1H, CH^Imidazole^), 7.65 (ddd, *J* = 8.7, 7.3, 1.6 Hz, 1H, CH^Ar^), 7.85 (d, *J* = 9.1 Hz, 1H, CH^Ar^), 8.15 (dd, *J* = 8.0, 1.5 Hz, 1H, CH^Ar^) ppm; ^13^C NMR (125.76 MHz, CDCl_3_) *δ* 112.8, 117.9, 118.5 (d, *J* = 23 Hz, C^ArF^), 119.0, 119.1, 120.0, 123.4, 125.7–125.8 (m, C^ArF^, C^Ar^), 126.1, 129.6 (d, *J* = 4 Hz, C^ArF^), 131.0, 131.1, 131.5, 132.8 (d, *J* = 2 Hz, C^ArF^), 132.9 (d, *J* = 9 Hz, C^ArF^), 134.7, 136.5, 137.0 (d, *J* = 2 Hz, C^Imidazole^), 137.4, 155.9, 156.2 (d, *J* = 259 Hz, C^ArF^), 158.6 (d, *J* = 2Hz, C^Pyranone^), 177.1 ppm; ^19^F NMR (376.44 MHz, CDCl_3_) *δ* 47.37 (d, *J* = 9.2 Hz, 1F) ppm; Anal. calcd. for C_24_H_15_FN_6_O_2_: C 65.75, H 3.45, N 19.17, found: C 65.95, H 3.32, N 18.84.

2-[3,5-Difluoro-4-(1*H*-1,2,4-triazol-1-yl)-2-methoxyphenyl]-4*H*-chromen-4-one (**18**). Yield 35 mg (20% according to technique A), 123 mg (69% according to technique B); white powder; mp 172–174 °C; IR *ν* 3143, 3108, 3063 (C–H^Ar^, C–H^Alk^), 1665 (C=O), 1523, 1464, 1377 (C=C^Ar^, C–H^Ar^, C–N), 1034 (C–F) cm^−1^; ^1^H NMR (400.13 MHz, CDCl_3_) *δ* 4.06 (d, *J* = 1.9 Hz, 3H, OCH_3_), 7.15 (s, 1H, CH^Pyranone^), 7.45–7.49 (m, 1H, CH^Ar^), 7.56 (dd, *J* = 8.4, 0.5 Hz, CH^Ar^), 7.66 (dd, *J* = 10.4, 2.3 Hz, CH^Ar^), 7.75 (ddd, *J* = 8.7, 7.2, 1.7 Hz, CH^Ar^), 8.25 (s, 1H, CH^Triazole^), 8.26 (dd, *J* = 7.9, 1.6 Hz, CH^Ar^), 8.45 (br.s, 1H, CH^Triazole^) ppm; ^13^C NMR (125.76 MHz, CDCl_3_) *δ* 62.1 (d, *J* = 6 Hz, C^OMe^), 111.2 (dd, *J* = 23, 4 Hz, C^ArF^), 113.6, 117.6 (dd, *J* = 16, 14 Hz, C^ArF^), 118.0, 123.8, 125.6, 125.8, 127.3 (dd, *J* = 9, 4 Hz, C^ArF^), 134.2, 144.0 (dd, *J* = 12, 4 Hz, C^ArF^), 145.4 (br.s, C^Triazole^), 150.5 (dd, *J* = 257, 4 Hz, C^ArF^), 151.4 (dd, *J* = 252, 3 Hz, C^ArF^), 153.1, 156.2, 157.2 (dd, *J* = 4, 2 Hz, C^Pyranone^), 178.2 ppm; ^19^F NMR (376.44 MHz, CDCl_3_) *δ* 29.29–29.30 (m, 1F), 37.19 (dd, *J* = 10.5, 1.3 Hz, 1F) ppm; Anal. calcd. for C_18_H_11_F_2_N_3_O_3_: C 60.85, H 3.12, N 11.83, found: C 60.92, H 3.26, N 11.72.

2-[5-Fluoro-2-methoxy-3,4-di(1*H*-1,2,4-triazol-1-yl)phenyl]-4*H*-chromen-4-one (**19**). Yield 24 mg (12%); white powder; mp 203–204 °C; IR *ν* 3113, 3100, 2953, 2924 (C–H^Ar^, C–H^Alk^), 1640 (C=O), 1510, 1467, 1374 (C=C^Ar^, C–H^Ar^, C–N), 1008 (C–F) cm^−1^; ^1^H NMR (400.13 MHz, CDCl_3_) *δ* 3.49 (s, 3H, OCH_3_), 7.16 (s, 1H, CH^Pyranone^), 7.48–7.52 (m, 1H, CH^Ar^), 7.59 (d, *J* = 8.4 Hz, 1H, CH^Ar^), 7.76–7.80 (m, 1H, CH^Ar^), 7.96–8.01 (m, 3H, CH^Triazole^), 8.27 (dd, *J* = 8.0, 1.4 Hz, 1H, CH^Ar^), 8.37 (d, *J* = 1.5 Hz, 1H, CH^Ar^), 8.42 (s, 1H, CH^Triazole^) ppm; ^13^C NMR (125.76 MHz, CDCl_3_) *δ* 62.4 (s, C^OMe^), 113.4, 118.0 (d, *J* = 24 Hz, C^ArF^), 118.0, 123.8, 124.9 (d, *J* = 15 Hz, C^ArF^), 125.9, 126.0, 129.0 (d, *J* = 8 Hz, C^ArF^), 129.6 (d, *J* = 2 Hz, C^Triazole^), 134.5, 145.7 (d, *J* = 2 Hz, C^Triazole^), 146.3, 151.0 (d, *J* = 4 Hz, C^ArF^), 152.6 (d, *J* = 253 Hz, C^ArF^), 152.8, 152.9, 156.3, 157.1 (d, *J* = 2 Hz, C^Pyranone^), 177.9 ppm; ^19^F NMR (376.44 MHz, CDCl_3_) *δ* 38.81 (dd, *J* = 9.9, 1.6 Hz, 1F) ppm; Anal. calcd. for C_20_H_13_FN_6_O_3_: C 59.41, H 3.24, N 20.78, found: C 59.51, H 3.23, N 20.79.

2-[5-Fluoro-2-hydroxy-3,4-di(1*H*-1,2,4-triazol-1-yl)phenyl]-4*H*-chromen-4-one (**22**). Yield 22 mg (11%); yellow powder; mp 313–316 °C; ^1^H NMR (400.13 MHz, (CD_3_)_2_SO) *δ* 7.04 (s, 1H, CH^Pyranone^), 7.52–7.56 (m, 1H, CH^Ar^), 7.78–7.80 (m, 1H, CH^Ar^), 7.88 (ddd, *J* = 8.6, 7.2, 1.6 Hz, 1H, CH^Ar^), 8.07 (s, 1H, CH^Triazole^), 8.09 (dd, *J* = 8.1, 1.5 Hz, 1H, CH^Ar^), 8.11 (s, 1H, CH^Triazole^), 8.26 (d, *J* = 10.4 Hz, 1H, CH^Ar^), 8.76 (s, 1H, CH^Triazole^), 8.79 (s, 1H, CH^Triazole^), 10.98 (s, 1H, OH) ppm; ^13^C NMR (125.76 MHz, (CD_3_)_2_SO) *δ* 112.6, 118.3 (d, *J* = 23 Hz, C^ArF^), 118.7, 123.3 (d, *J* = 9 Hz, C^ArF^), 124.0, 124. 7, 124.8 (d, *J* = 15 Hz, C^ArF^), 125.7, 134.5, 146.6, 147.3, 148.7 (d, *J* = 2 Hz, C^Triazole^), 149.1 (d, *J* = 244 Hz, C^ArF^), 152.3, 152.5, 156.0, 159.0, 177.1 ppm; ^19^F NMR (376.44 MHz, (CD_3_)SO) *δ* 30.74 (d, *J* = 9.6 Hz, 1F) ppm.

2-[3,5-Difluoro-4-(1*H*-imidazol-1-yl)-2-methoxyphenyl]-4*H*-chromen-4-one (**23**). Yield 39 mg (23% according to technique A), 94 mg (53% according to technique B); white powder; mp 176–177 °C; IR *ν* 3108, 3043, 2999, 2950 (OMe, C–H^Ar^), 1630 (C=O), 1524, 1480, 1370 (C=C^Ar^, C–H^Ar^, C–O, C–N), 1112 (C–F) cm^−1^; °C; ^1^H NMR (400.13 MHz, CDCl_3_) *δ* 4.04 (d, *J* = 1.7 Hz, 3H, OMe), 7.14 (s, 1H, CH^Pyranone^), 7.27–7.29 (m, 2H, 2CH^Imidazole^), 7.45–7.49 (m, 1H, 1CH^Ar^), 7.55–7.57 (m, 1H, 1CH^Ar^), 7.64 (dd, *J* = 10.8, 2.2 Hz, 1H, CH^Ar^), 7.75 (ddd, *J* = 8.6, 7.2, 1.7 Hz, 1H, CH^Ar^), 7.83 (br.s., 1H, CH^Imidazole^), 8.25 (dd, *J* = 8.0, 1.5 Hz, 1H, CH^Ar^) ppm; ^13^C NMR (125.76 MHz, CDCl_3_) *δ* 62.1 (d, *J* = 6 Hz, C^OMe^), 111.2 (dd, *J* = 24, 4 Hz, C^Ar^), 113.3, 118.0, 118.2 (dd, *J* = 14, 2 Hz, C^Ar^), 120.0 (t, *J* = 2 Hz, C^Imidazole^), 123.8, 125.5–125.6 (m, C^Ar^,C^Imidazole^), 125.8, 129.9, 134.2, 137.7 (t, *J* = 3 Hz, C^Imidazole^), 144.2 (dd, *J* = 12, 4 Hz, C^Ar^), 150.1 (dd, *J* = 254, 4 Hz, C^ArF^), 151.1 (dd, *J* = 249, 4 Hz, C^ArF^), 156.2, 157.4 (dd, *J* = 4, 2 Hz, C^Pyranone^), 178.2 ppm; ^19^F NMR (376.44 MHz, CDCl_3_) *δ* 27.59–27.60 (m, 1F), 36.45–36.48 (m, 1F)ppm; Anal. calcd. for C_19_H_12_F_2_N_2_O_3_: C 64.41, H 3.41, N 7.91, found: C 64.37, H 3.40, N 8.02.

2-[2,5-Difluoro-3,6-di(1*H*-pyrazol-1-yl)-4-(1*H*-1,2,4-triazol-1-yl)phenyl]-4*H*-chromen-4-one (**26**). Yield 39 mg (17%); white powder; mp 256–258 °C; IR *ν* 3130, 3106, 3071 (C–H^Ar^), 1642 (C=O), 1529, 1483, 1390 (C=C^Ar^, C–H^Ar^, C–N), 1139, 1127 (C–F) cm^−1^; ^1^H NMR (400.13 MHz, CDCl_3_) *δ* 6.44 (d, *J* = 1.2 Hz, 1H, C^Pyranone^), 6.45–6.46 (m, 1H, CH^Imidazole^), 6.48–6.49 (m, 1H, CH^Imidazole^), 7.19–7.21 (m, 1H, CH^Ar^), 7.40–7.44 (m, 1H, CH^Ar^), 7.58 (d, *J* = 1.6 Hz, 1H, C^Ar^), 7.63–7.66 (m, 3H, CH^Imidazole^), 7.84 (t, *J* = 2.6 Hz, 1H, C^Imidazole^), 8.05 (s, 1H, CH^Triazole^), 8.19 (dd, *J* = 8.0, 1.5 Hz, 1H, C^Ar^), 8.24 (s, 1H, CH^Triazole^) ppm; ^13^C NMR (125.76 MHz, CDCl_3_) *δ* 108.4, 108.6, 115.0 (d, *J* = 3 Hz, C^Pyrazole^), 118.0, 121.4 (d, *J* = 17 Hz, C^ArF^), 123.7, 125.5 (dd, *J* = 14, 3 Hz, C^ArF^), 125.6, 125.8, 126.3 (d, *J* = 17 Hz, C^ArF^), 128.7 (dd, *J* = 14, 3 Hz, C^ArF^), 132.0 (d, *J* = 4 Hz, C^Pyrazole^), 132.3 (d, *J* = 2 Hz, C^Pyrazole^), 134.1, 142.7, 142.9, 145.7 (d, *J* = 1 Hz, C^Pyranone^), 148.0 (dd, *J* = 258, 4 Hz, C^ArF^), 151.1 (dd, *J* = 257, 4 Hz, C^ArF^), 153.1, 154.3 (d, *J* = 2 Hz, C^Pyranone^), 177.2 ppm; ^19^F NMR (376.44 MHz, CDCl_3_) *δ* 30.62 (dd, *J* = 14.4, 1.8 Hz, 1F), 30.75–39.80 (m, 1F) ppm; Anal. calcd. for C_23_H_13_F_2_N_7_O_2_: C 60.40, 2.86, N 21.44, found: C 60.15, H 2.68, N 21.54.

2-[2,3,5,6-Tetra(1*H*-pyrazol-1-yl)-4-(1*H*-1,2,4-triazol-1-yl)phenyl]-4*H*-chromen-4-one (**27**). Yield 136 mg (49%); yellow powder; mp 299–301 °C; IR *ν* 3126, 3092 (C–H^Ar^), 1646 (C=O), 1525, 1468, 1389 (C=C^Ar^, C–H^Ar^, C–N) cm^−1^; ^1^H NMR (400.13 MHz, CDCl_3_) *δ* 5.30 (s, 1H, CH^Pyrazole^), 6.08 (s, 1H, CH^Pyrazole^), 6.16–6.18 (m, 5H, CH^Pyranone^, CH^Pyrazole^), 7.11 (d, *J* = 8.4 Hz, 1H, CH^Ar^), 7.24 (d, *J* = 2.5 Hz, 1H, CH^Pyrazole^), 7.29 (d, *J* = 2.5 Hz, 1H, CH^Pyrazole^), 7.32–7.36 (m, 1H, CH^Ar^), 7.43 (d, *J* = 1.6 Hz, 2H, CH^Pyrazole^), 7.49 (d, *J* = 1.6 Hz, 2H, CH^Pyrazole^), 7.54–7.59 (m, 1H, CH^Ar^), 7.74 (s, 1H, CH^Triazole^), 8.05 (s, 1H, CH^Triazole^), 8.06–8.08 (m, 1H, CH^Ar^) ppm; ^13^C NMR (125.76 MHz, CDCl_3_) *δ* 108.0 (s, 2C^Pyrazole^), 108.2 (s, 2C^Pyrazole^), 113.3, 117.8, 123.3, 125.4, 125.7 (s, 2C^Ar^), 131.8 (s, 2C^Pyrazole^), 132.1 (s, 2C^Pyrazole^), 132.7, 133.8 (s, 2C^Ar^), 135.2, 135.9, 138.2, 142.2 (s, 2C^Pyrazole^), 142.3 (s, 2C^Pyrazole^), 145.9, 152.3, 156.0, 156.6, 177.0 ppm; Anal. calcd. for C_29_H_19_N_11_O_2_: C 62.92, 3.46, N 27.83, found: C 63.15, H 3.68, N 27.69.

2-[4-(1*H*-Imidazol-1-yl)-2,3,5,6-tetra(1*H*-pyrazol-1-yl)phenyl]-4*H*-chromen-4-one (**28**). Yield 31 mg (11%); yellow powder; mp 320–321 °C; IR *ν* 3127 (C–H^Ar^), 1649 (C=O), 1525, 1467, 1388 (C=C^Ar^, C–H^Ar^, C–N) cm^−1^; ^1^H NMR (400.13 MHz, CDCl_3_) *δ* 6.03 (s, 1H, CH^Pyranone^), 6.12–6.13 (m, 1H, CH^Pyrazole^), 6.15 (dd, *J* = 4.3, 2.2 Hz, 4H, CH^Pyrazole^), 7.11 (d, *J* = 8.3 Hz, 1H, CH^Ar^), 7.20–7.23 (m, 3H, CH^Pyrazole^, CH^Imidazole^), 7.31–7.34 (m, 3H, CH^Ar^, CH^Pyrazole^, CH^Imidazole^), 7.39 (d, *J* = 1.6 Hz, 2H, CH^Pyrazole^), 7.41 (d, *J* = 1.6 Hz, 1H, CH^Imidazole^), 7.47 (d, *J* = 1.5 Hz, 2H, CH^Pyrazole^), 7.55 (ddd, *J* = 8.6, 7.3, 1.6 Hz, 1H, CH^Ar^), 8.06 (dd, *J* = 7.9, 1.5 Hz, 1H, CH^Ar^) ppm; ^13^C NMR (125.76 MHz, CDCl_3_) *δ* 107.3, 107.5 (s, 2C^Pyrazole^), 107.8 (s, 2C^Pyrazole^), 113.2, 117.9, 123.3, 125.3, 125.6, 131.5, 131.9 (s, 2C^Ar^), 132.0 (s, 2C^Pyrazole^), 132.2 (s, 2C^Pyrazole^), 133.7, 135.8, 138.3, 138.4, 141.7 (s, 2C^Ar^), 141.7 (s, 2C^Pyrazole^), 141.9 (s, 2C^Pyrazole^), 156.0, 157.1, 177.1 ppm; Anal. calcd. for C_30_H_20_N_10_O_2_: C 65.21, 3.65, N 25.35, found: C 65.15, H 3.57, N 25.55.

2-[4-(1*H*-Imidazol-1-yl)-2,3,5,6-tetra(1*H*-1,2,4-triazol-1-yl)phenyl]-4*H*-chromen-4-one (**29**). Yield 92 mg (33%); pale pink powder; mp 299–301 °C; IR *ν* 3123, 3105 (C–H^Ar^), 1647 (C=O), 1510, 1464, 1378 (C=C^Ar^, C–H^Ar^, C–N) cm^−1^; ^1^H NMR (500.13 MHz, (CD_3_)_2_SO) *δ* 6.30 (s, 1H, CH^Pyranone^), 6.84 (s, 1H, CH^Imidazole^), 7.01 (t, *J* = 1.3 Hz, 1H, CH^Imidazole^), 7.34 (d, *J* = 8.2 Hz, 1H, CH^Ar^), 7.47–7.49 (m, 1H, CH^Ar^), 7.52 (s, 1H, CH^Imidazole^), 7.80 (ddd, *J* = 8.7, 7.2, 1.7 Hz, 1H, CH^Ar^), 7.92 (dd, *J* = 8.0, 1.6 Hz, 1H, CH^Ar^), 8.08 (s, 2H, CH^Triazole^), 8.11 (s, 2H, CH^Triazole^), 8.63 (s, 2H, CH^Triazole^), 8.82 (s, 2H, CH^Triazole^) ppm; ^13^C NMR (125.76 MHz, CDCl_3_) *δ* 113.7, 117.9, 121.0, 122.4, 124.9, 126.3, 129.3, 130.6, 134.0 (s, 2C^Ar^), 135.0, 135.5 (s, 2C^Ar^), 136.0, 137.9, 146.7 (s, 2C^Triazole^), 146.8 (s, 2C^Triazole^), 152.9 (s, 2C^Triazole^), 152.9 (s, 2C^Triazole^), 154.8, 155.4, 175.4 ppm; Anal. calcd. for C_26_H_16_N_14_O_2_: C 56.11, 2.90, N 35.24, found: C 56.30, H 2.98, N 35.04.

2-[4-(1*H*-Pyrazol-1-yl)-2,3,5,6-tetra(1*H*-1,2,4-triazol-1-yl)phenyl]-4*H*-chromen-4-one (**31**). Yield 159 mg (59%); pale pink powder; mp 309–310 °C; IR *ν* 3113 (C–H^Ar^), 1651 (C=O), 1511, 1466, 1386 (C=C^Ar^, C–H^Ar^, C–N) cm^−1^; ^1^H NMR (500.13 MHz, (CD_3_)_2_SO) *δ* 6.31–6.32 (m, 1H, CH^Pyrazole^), 6.32 (s, 1H, CH^Pyranone^), 7.34 (d, *J* = 8.2 Hz, 1H, CH^Ar^), 7.46–7.49 (m, 1H, CH^Ar^), 7.56 (d, *J* = 1.7 Hz, 1H, CH^Pyrazole^), 7.67 (d, *J* = 2.5 Hz, 1H, CH^Pyrazole^), 7.79 (ddd, *J* = 8.7, 7.2, 1.7 Hz, 1H, CH^Ar^), 7.92 (dd, *J* = 8.0, 1.6 Hz, 1H, CH^Ar^), 8.04 (s, 2H, CH^Triazole^), 8.05 (s, 2H, CH^Triazole^), 8.53 (s, 2H, CH^Triazole^), 8.82 (s, 2H, CH^Triazole^) ppm; ^13^C NMR (125.76 MHz, (CD_3_)_2_SO) *δ* 107.9, 113.7, 118.0, 122.4, 124.9, 126.3, 130.8, 133.1, 133.9 (s, 2C^Ar^), 135.1, 135.4 (s, 2C^Ar^), 137.6, 142.5, 146.8 (s, 2C^Triazole^), 146.9 (s, 2C^Triazole^), 152.7 (s, 2C^Triazole^), 152.9 (s, 2C^Triazole^), 155.0, 155.4, 175.5 ppm; Anal. calcd. for C_26_H_16_N_14_O_2_: C 56.11, 2.90, N 35.24, found: C 56.26, H 2.99, N 35.11.

### 3.3. XRD Experiments

The X-ray studies were performed on an Xcalibur 3 CCD (Oxford Diffraction Ltd., Abingdon, UK) diffractometer with a graphite monochromator, λ(MoKα) 0.71073 Å radiation and T 295(2) K. An empirical absorption correction was applied. Using Olex2 [51], the structure was solved with the Superflip [52] structure solution program using charge flipping and refined with the ShelXL [53] refinement package using Least Squares minimization. All non-hydrogen atoms were refined in the anisotropic approximation; H-atoms at the C–H bonds were refined in the “rider” model with dependent displacement parameters. An empirical absorption correction was carried out through spherical harmonics, implemented in the SCALE3 ABSPACK scaling algorithm by the program “CrysAlisPro” (Rigaku Oxford Diffraction).

The main crystallographic data for **4**: C_17_H_7_F_4_N_3_O_2_, *M* 361.26, orthorhombic, *a* 15.8944(12), *b* 12.7694(11), *c* 7.3245(6) Å, *V* 1486.6(2) Å^3^, space group Pna2_1_, *Z* 4, μ(Mo Kα) 0.125 mm^–1^, 256 refinement parameters, 3609 reflections measured, and 2493 unique (*R*_int_ = 0.0617), which were used in all calculations. CCDC 2225826 contains the supplementary crystallographic data for this compound.

The main crystallographic data for **8**: C_18_H_8_F_4_N_2_O_2_, *M* 360.26, monoclinic, *a* 15.0164(11), *b* 7.9494(7), *c* 12.8592(10) Å, *β* 99.256(7)°, *V* 1515.0(2) Å^3^, space group P2_1_/c, *Z* 4, μ(Mo Kα) 0.125 mm^–1^, 268 refinement parameters, 4162 reflections measured, and 2246 unique (*R*_int_ = 0.0633), which were used in all calculations. CCDC 2,225,827 contains the supplementary crystallographic data for this compound.

The main crystallographic data for **15**: C_18_H_9_F_3_N_2_O_2_, *M* 342.27, monoclinic, *a* 13.3565(10), *b* 7.8477(5), *c* 14.7075(12) Å, *β* 113.251(9)°, *V* 1416.4(2) Å^3^, space group P2_1_/n, *Z* 4, μ(Mo Kα) 0.125 mm^–1^, 262 refinement parameters, 3865 reflections measured, and 2692 unique (*R*_int_ = 0.0597), which were used in all calculations. CCDC 2,225,828 contains the supplementary crystallographic data for this compound.

The main crystallographic data for **19**: C_20_H_13_FN_6_O_3_, *M* 404.36, triclinic, *a* 13.3565(10), *b* 7.8477(5), *c* 14.7075(12) Å, *α* 111.535(14), *β* 94.311(13), *γ* 101.586(12)°, *V* 1416.4(2) Å^3^, space group P$\bar{1}$, *Z* 2, μ(Mo Kα) 0.125 mm^–1^, 288 refinement parameters, 3662 reflections measured, 1504 unique (*R*_int_ = 0.0711) which were used in all calculations. CCDC 2,225,829 contains the supplementary crystallographic data for this compound.

The main crystallographic data for **23**: C_19_H_12_F_2_N_2_O_3_, *M* 354,31, monoclinic, *a* 14.0716(10), *b* 7.7151(5), *c* 14.5964(12) Å, *β* 100.453(7)°, *V* 1553.0(2) Å^3^, space group P2_1_/c, *Z* 4, μ(Mo Kα) 0.125 mm^–1^, 257 refinement parameters, 4236 reflections measured, and 2349 unique (*R*_int_ = 0.0568) which were used in all calculations. CCDC 2,225,830 contains the supplementary crystallographic data for this compound. 

### 3.4. Fungistatic Activity Evaluation

The following dermatophyte fungal strains were used: *Trichophyton rubrum* (RCPF F 1408), *Epidermophyton floccosum* (RCPF F 1659/17), and *Microsporum canis* (RCPF F 1643/1585), as well as yeast-like fungus *Candida parapsilosis* (RCPF 1245/ ATCC 22019). The fungi cultures were obtained from the Russian Collection of Pathogenic Fungi (Kashkin Research Institute of Medical Mycology; Mechnikov Northwest State Medical University, St.-Petersburg). Saburo agar and Saburo broth were used for the fungi. The microorganisms were identified as matrix-extracted bacterial proteins with an accuracy of 99.9% using a BioMerieux VITEK MS MALDI-TOF analyzer. The test cultures were prepared to an optical density of 0.5 according to McFarland (1.5 × 108 CFU/ mL) using a BioMerieux DensiCHEK densimeter. The suspensions of *C. parapsilosis* were prepared from 24 h cultures, and dermatophyte inocula were prepared after incubation for 2 weeks and preliminary homogenization in sterile saline. The fungi were inoculated at a concentration of 10^5^ CFU/mL. The antimycotic activity was evaluated by a micro method [54]. The agar nutrient medium was maintained in liquid by heating to 52 °C. The chemical compounds to be tested were dissolved in DMSO to a concentration of 1000 μg/ mL, and the stock solutions were diluted with distilled sterilized water; serial dilutions (from 250–200 μg/ mL) were made using nutrient media. Dermatophytes were incubated at 27 °C for up to 7–10 days and *C. parapsilosis* for 24 h in a moist 5.0% CO_2_ chamber. In each case, positive and negative controls were used. The minimum inhibitory concentration was determined visually as the lowest concentration at which a test culture no longer grows. Chemically pure fluconazole was used as a reference drug.

## 4. Conclusions

The data obtained in this work and earlier in the study of transformations with pyrazole [29] thus indicate that base-promoted reactions of nucleophilic aromatic substitution are a convenient method for the functionalization of polyfluoroflavones **1**–**3** with azoles with different numbers of nitrogen atoms. At the same time, it was found that monosubstitution of the para-fluorine atom successfully and selectively occurs while using the system (azole (1.5 equiv.)/NaOBu*^t^* (1.5 equiv.)/MeCN) regardless of the structure and properties of the used polyfluorinated substrates and nucleophilic reagents, since in all cases, mono(azolyl)-substituted flavones were obtained in good yields. Under the conditions (azole (6 equiv.)/NaOBu*^t^* (6 equiv.)/DMF), which promote the formation of persubstituted products, the interactions of polyfluoroflavones **1**–**3** with pyrazole are distinguished by high selectivity [26], while similar reactions with triazole produced productively only for penta- and tetrafluoroflavones **1** and **2**, and the same transformations with imidazole in general are extremely non-selective.

Comparing the conversion of polyfluoroflavones **1**–**3** in reactions with azoles under conditions that do not provide selective substitution (azole (3 equiv.)/Cs_2_CO_3_ (3 equiv.)/MeCN)), it can be noted that transformations with pyrazole [26] are characterized by easier formation of polysubstituted products, and under conditions conducive to persubstitution, it is possible to build the following series of azoles according to reactivity: pyrazole ≥ triazole > imidazole. Obviously, in both cases, the reactivity of azoles does not correspond to their basicity, and therefore to some extent, nucleophilicity, since imidazole is known to be the strongest base among them [55]. According to the literature data [56,57], polyfluoroaromatic compounds generally react with nucleophiles via Meisenheimer complexes. For flavones **1**–**3,** we assume an analogous mechanism [26]; first nucleophile attack occurs on the activated C4′ site of flavones **1**–**3** with the generation of an intermediate of a stable quinoid structure, followed by formation of a mono(azolyl)-substituted product. Sequential substitution is coordinated by the joint activating effect of the substituents, and the resulting intermediate complexes are stabilized both by O,N-bidentate coordination between the azole and the pyrone fragment of the molecule with Na^+^ and by the coordination of neighboring azole moieties with Na^+^. It is likely that the higher reactivity of pyrazole, triazole, and their intermediates compared to imidazole and its derivatives in S_N_Ar poly- and per-substitution reactions is due to the possibility of participation of their imine nitrogen atoms in the N–N=C function in coordination during the formation of transitional complexes.

In addition, it was shown that the per-substitution conditions can be successfully used for the synthesis of polynuclear hybrid compounds containing two different azole fragments by the reaction of mono(azolyl)-substituted flavones with pyrazole and triazole.

Using XRD data, the structural features of triazolyl- and imidazolyl-substituted flavones in crystal form were established. For example, in contrast to previously synthesized pyrazole analogues [26], new azole derivatives do not contain an intramolecular H-bond.

In terms of possible practical applications, it has been established that the resulting poly(pyrazolyl)-substituted flavones have luminescent properties, which makes further development of research in this area promising. In addition, weak antimycotic activity was found for some azolyl-containing flavones.

## Data Availability

Not applicable.

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
