# Peer review of "Synthesis of Mono- and Polyazole Hybrids Based on Polyfluoroflavones"

_molecules, 2023, doi:10.3390/molecules28020869_

Round 1
Reviewer 1 Report
In the manuscript the Authors present studies on synthesis and properties of azole substituted flavones, via SNAr reaction of 2-(polyfluorophenyl)-chromenones. The area is a continuation of their previous reports on synthesis of precursors and related reactions with pyrazole, instead of imidazole and triazole, presented in the current paper (J. Fluorine Chem. 2020, 240, 109657, J. Fluorine Chem. 2021, 249, 109857 and J. Fluorine Chem. 2022, 263, 110034). From this perspective originality/novelty is rather moderate, but at the same time the studies are well documented, comprehensive, and shed new light on multiple SNAr substitution processes in polyfluoroarenes. In particular, the Authors present in detail contents of mixtures produced in reactions carried out with Cs2CO3, and differentiate reaction output, depending on the reaction conditions.
There is only a limited number of lapses/errors, which should be corrected prior publication - more formal issues are listed at the end of my comment. Sentence in lines 132-133 states: 'Compound 20 then undergoes a nucleophilic addition reaction with triazole, forming flavone 21.' This is puzzling, as OH group can deprotonate under reaction conditions, and such phenolic anion is by no means a leaving group in SNAr (or any other reaction). Rather demethylation of OMe forms 20, and attack of triazole anion at 3 displaces OMe(-), giving 21.
Also fact that only mono-substituted (para) products are characterized by X-ray is disappointing, poly-azole hybrids, as e.g. structure of compound 31, would be much more interesting. Solubility issues, mentioned in the text, could be solved by use of alkyl (e.g tert-butyl) substituted azoles.
In the Experimental section (lines 279, and 290) information on drying of organic extracts prior chromatography is missing. Was is, in fact, the case? MORE IMPORTANTLY, C6F6 is given as internal standard for 19F NMR (line 269), but at spectra reproductions main peak is assigned to 0.0 ppm, instead of -164.9 ppm. This causes huge changes of chemical shifts listed in the characterization data. E.g. typical chemical shifts of CAr-F in the products can be around -140 ppm, but in this work they appear at ca. +30 ppm (difference equals to -164.9 ppm). For discussion of acceptable chemical shift errors in 19F NMR spectroscopy (around 1 ppm), see: Angew. Chem. Int. Ed. 2018, 57, 9528. For data, see, e.g.: https://www.colorado.edu/lab/nmr/sites/default/files/attached-files/19f_nmr_reference_standards_0.pdf
FORMAL ISSUES:
line 15, abstract, is 'The ability of penta- and tetrafluoroflavones to form perfluoro-substituted products with triazole...', why 'perfluoro-substituted'? Rather 'persubstituted' or 'perazole-substituted' (it should refer to the type of product, rather than type of the running process). The same applies to line 227 in the main text.
Figures 2 and 3 - I would move unit cell presentations (2b, 3b) at the X-ray structures to the SI (it is not very informative), but add projection of 4 and 8 at different orientation, i.e. along longest axis of the molecule,
lines 132-133, the sentence suggests that 2,3,4,5-tetrafluorophenyl substitution imidazole (rewrite, please),
There is no Table 3 in the text, so numbering should be corrected.
lines 553, 562 and 584, reference should be [26], not [29]?
Author Response
In the manuscript the Authors present studies on synthesis and properties of azole substituted flavones, via SNAr reaction of 2-(polyfluorophenyl)-chromenones. The area is a continuation of their previous reports on synthesis of precursors and related reactions with pyrazole, instead of imidazole and triazole, presented in the current paper (J. Fluorine Chem. 2020, 240, 109657, J. Fluorine Chem. 2021, 249, 109857 and J. Fluorine Chem. 2022, 263, 110034). From this perspective originality/novelty is rather moderate, but at the same time the studies are well documented, comprehensive, and shed new light on multiple SNAr substitution processes in polyfluoroarenes. In particular, the Authors present in detail contents of mixtures produced in reactions carried out with Cs2CO3, and differentiate reaction output, depending on the reaction conditions.
Authors: We appreciate the time and effort that the reviewer have dedicated to providing valuable feedback on our manuscript. And we thank the reviewer for the positive evaluation of our manuscript.
R.: There is only a limited number of lapses/errors, which should be corrected prior publication - more formal issues are listed at the end of my comment. Sentence in lines 132-133 states: 'Compound 20 then undergoes a nucleophilic addition reaction with triazole, forming flavone 21.' This is puzzling, as OH group can deprotonate under reaction conditions, and such phenolic anion is by no means a leaving group in SNAr (or any other reaction). Rather demethylation of OMe forms 20, and attack of triazole anion at 3 displaces OMe(-), giving 21.
Authors: Our assumption about the formation of product 20 is based on our previous experience, which we obtained in the study of SNAr reactions of flavone 3 with pyrazole. Using column chromatography, we isolated the product from this reaction, which, according to HPLC-MS and 1H NMR spectroscopy, was an inseparable mixture of 2-[2,3,4,5-tetra(1H-pyrazol-1-yl)phenyl]-4H-chromen-4-one and its intermediate, 2-[2-hydroxy-3,4,5-tri(1H-pyrazol-1-yl)phenyl]-4H-chromen-4-one, in the ratio 6.6÷1 [Shcherbakov, K.V.; Panova, M.A.; Burgart, Y.V.; Saloutin, V.I. Selective nucleophilic aromatic substitution of 2-(polyfluorophenyl)-4H-chromen-4-ones with pyrazole. J. Fluorine Chem. 2022, 263, 110034. https://doi.org/10.1016/j.jfluchem.2022.110034]. Further reaction of this mixture with pyrazole in the presence of NaOBut in DMF resulted in the formation of a tetrapyrazolyl-substituted flavone. We explain such process by the fact that the formed phenol group can be in equilibrium between the keto and enol forms, and the keto form can undergo nucleophilic addition followed by aromatization to obtain a persubstitution product. We have added a paragraph explaining the possibility of the formation of compound 21 from 20 in the corrected version of the article (lines 170-172).
R.: Also fact that only mono-substituted (para) products are characterized by X-ray is disappointing, poly-azole hybrids, as e.g. structure of compound 31, would be much more interesting. Solubility issues, mentioned in the text, could be solved by use of alkyl (e.g tert-butyl) substituted azoles.
Authors: Unfortunately, in this work, we were indeed unable to grow crystals of poly- and per-substituted flavones due to their poor solubility in organic solvents. However, we are encouraged by the results on the luminescence properties of such compounds, so we plan to continue research in this area, including using alkyl (e.g tert-butyl) substituted azoles. We are grateful for the reviewer's suggestion, which is valuable for the development of this work. The present work is devoted to the study of interactions of polyfluoroflavones with unsubstituted azoles.
R.: In the Experimental section (lines 279, and 290) information on drying of organic extracts prior chromatography is missing. Was is, in fact, the case?
Authors: Organic extracts were evaporated and immobilized on silica gel prior chromatography. This information was added to the description of the synthetic procedure (lines 283 and 294).
R.: MORE IMPORTANTLY, C6F6 is given as internal standard for 19F NMR (line 269), but at spectra reproductions main peak is assigned to 0.0 ppm, instead of -164.9 ppm. This causes huge changes of chemical shifts listed in the characterization data. E.g. typical chemical shifts of CAr-F in the products can be around -140 ppm, but in this work they appear at ca. +30 ppm (difference equals to -164.9 ppm). For discussion of acceptable chemical shift errors in 19F NMR spectroscopy (around 1 ppm), see: Angew. Chem. Int. Ed. 2018, 57, 9528. For data, see, e.g.: https://www.colorado.edu/lab/nmr/sites/default/files/attached-files/19f_nmr_reference_standards_0.pdf
Authors: The internal standard was C6F6 for 19F NMR spectra and chemical shifts were not converted to CCl3F. According to the guidelines for the authors of the ”Molecules” there are no specific requirements for the presentation of 19F NMR spectra, but if it is important for the reviewer that the chemical shifts be presented relative to the CCl3F, we can recalculate. For now, the information that chemical shifts were not converted to CCl3F was added to the manuscript (line 272).
R.: FORMAL ISSUES:
R.: line 15, abstract, is 'The ability of penta- and tetrafluoroflavones to form perfluoro-substituted products with triazole...', why 'perfluo ro-substituted'? Rather 'persubstituted' or 'perazole-substituted' (it should refer to the type of product, rather than type of the running process). The same applies to line 227 in the main text.
Authors: It was corrected.
R.: Figures 2 and 3 - I would move unit cell presentations (2b, 3b) at the X-ray structures to the SI (it is not very informative), but add projection of 4 and 8 at different orientation, i.e. along longest axis of the molecule,
Authors: We have provided unit cell presentations at the X-ray structures (Figures 2b, 3b) to show that the presence of different azole moieties in flavones having the same amount of fluorine in the phenyl substituent significantly affects the packing in crystals (a cell of compound 4 has a rhombic syngony and compound 8 has a monoclinic). Therefore, we would be grateful to the editor and the reviewer if we would be allowed to dwell on our version of the of figures.
R.: lines 132-133, the sentence suggests that 2,3,4,5-tetrafluorophenyl substitution imidazole (rewrite, please),
Authors: It has been rewritten.
R.: There is no Table 3 in the text, so numbering should be corrected.
Authors: The numbering of tables has been changed.
R.: lines 553, 562 and 584, reference should be [26], not [29]?
Authors: The correct reference is [26], it has been changed.
Reviewer 2 Report
In their submission to Molecule entitled “Synthesis of mono- and polyazole hybrids based on polyfluoroflavones”, Saloutin and coworkers describe the preparation of polyazole derivatives. The products are well characterized and are of great interest to be used as materials for OLED applications. Thus, this paper has the quality to be published in a journal like Molecules, but some points need to be addressed.
A) Tables 1-6 should be moved to the Supporting Information.
B) The X-ray description is too long and need to be shortened.
C) The authors should comment on the reaction mechanism.
Thus, my recommendation is to accept this manuscript after minor revision.
Author Response
R.: In their submission to Molecule entitled “Synthesis of mono- and polyazole hybrids based on polyfluoroflavones”, Saloutin and coworkers describe the preparation of polyazole derivatives. The products are well characterized and are of great interest to be used as materials for OLED applications. Thus, this paper has the quality to be published in a journal like Molecules, but some points need to be addressed.
Authors: We thank the reviewer for the time contributed to our manuscript and for the positive assessment of our study.
R.: A) Tables 1-6 should be moved to the Supporting Information.
Author: Tables 1-6 are very important in this work because they contain information on the composition of the mixture of products formed when carrying out the reaction using Cs2CO3. In addition, they provide 19F NMR characteristics of these compounds, which were used for their identification. In some cases, these products have not been isolated individually, so there is no information about their structure in the experimental part. Without these tables, the data on transformations under the action of Cs2CO3 will look unfounded and not convincing.
R.: B) The X-ray description is too long and need to be shortened.
Authors: There are significant differences in the crystal structures and cell units of the molecules depending on the azole substituent and the amount of fluorine atoms in the phenyl moiety, that is why we would like to highlight these features. And we would be grateful to the editor and the reviewer if we would be allowed to keep the description of crystal structures unchanged.
R.: C) The authors should comment on the reaction mechanism.
Authors: The reaction mechanism of studied transformations are considered in detail by us in the article [Shcherbakov, K.V.; Panova, M.A.; Burgart, Y.V.; Saloutin, V.I. Selective nucleophilic aromatic substitution of 2-(polyfluorophenyl)-4H-chromen-4-ones with pyrazole. J. Fluorine Chem. 2022, 263, 110034. https://doi.org/10.1016/j.jfluchem.2022.110034], therefore, in this work we did not provide it, but only gave a link to this work. However, we appreciate the reviewer's recommendation and have added a few sentences to discuss this point in the conclusion: “For flavones 1-3 we assume analogous mechanism [26], first the nucleophile attack occurs on the activated C4′ site of flavones 1-3 with the generation of intermediate of a stable quinoid structure, followed by formation of monoazolyl-substituted product. Sequential substitution is coordinated by the joint activating effect of the substituents, and the resulting intermediate complexes are stabilized both by O,N-bidentant coordination between the azole and the pyrone fragment of the molecule with Na+, and by the coordination of neighboring azole moieties with Na+.” (lines 578-584).
Reviewer 3 Report
The authors present an extension of their previous work describing the reaction of pyrazole with polyfluorinated flavones. In the current work, they extend the reactivity towards the use of 1H-1,2,4-triazole and imidazole. Overall, this is a very clearly written manuscript and the experimental work appears to be carried out thoroughly and carefully. Including the x-ray crystal structures greatly strengthens the arguments made by the authors. I just have two minor suggestions:
1. It would be helpful to add a scheme to be able to better visualize what the authors are discussing about their previous work based on reference 29 (i.e., to depict what is described in lines 48-52.
2. On line 231, I assume that the authors were also unable to obtain a crystal suitable for XRD for compound 26 as well?
Upon consideration of the above two suggestions, I recommend publication of this work.
Author Response
R.: The authors present an extension of their previous work describing the reaction of pyrazole with polyfluorinated flavones. In the current work, they extend the reactivity towards the use of 1H-1,2,4-triazole and imidazole. Overall, this is a very clearly written manuscript and the experimental work appears to be carried out thoroughly and carefully. Including the x-ray crystal structures greatly strengthens the arguments made by the authors. I just have two minor suggestions:
R.: 1. It would be helpful to add a scheme to be able to better visualize what the authors are discussing about their previous work based on reference 29 (i.e., to depict what is described in lines 48-52.
Authors: We would like to clarify which reference the reviewer is referring to: 26 - which is about the transformations of polyfluoroflavones with pyrazole, or 29 - which is about the synthesis of the initial flavones? From our point of view, adding additional graphic material is unnecessary, so we have only provided links to previous studies.
R.; 2. On line 231, I assume that the authors were also unable to obtain a crystal suitable for XRD for compound 26 as well?
Authors: Unfortunately, in this work, we were also unable to grow suitable crystal of compound 26 for XRD analysis.
R.: Upon consideration of the above two suggestions, I recommend publication of this work.
Authors: We thank the reviewer for the positive evaluation of our manuscript.